# EvoFlows: Evolutionary Edit-Based Flow-Matching for Protein Engineering

**Nicolas Deutschmann**[*] , **Constance Ferragu**[*], **Jonathan D. Ziegler**[*], **Shayan Aziznejad & Eli Bixby**
Cradle
Zürich, CH
`{nicolas, constance, jonathan}@cradle.bio`

## Abstract

We introduce EvoFlows, a variable-length protein sequence-to-sequence modeling approach designed for protein engineering. Existing protein language models are poorly suited for optimization tasks: autoregressive models require full sequence generation, masked language and discrete diffusion models rely on pre-specified mutation locations, and no existing methods naturally support insertions and deletions relative to a template sequence. EvoFlows learns mutational trajectories between evolutionarily related protein sequences via edit flows, allowing it to perform a controllable number of mutations (insertions, deletions, and substitutions) on a template sequence, predicting not only *which* mutation to perform, but also *where* it should occur. Through extensive *in silico* evaluation on diverse protein families from UniRef and OAS, we show that EvoFlows generates variants that remain consistent with natural protein families while exploring farther from template sequences than leading baselines.

## 1 Introduction

Protein optimization aims to improve the properties of an existing protein through targeted mutations (substitutions, insertions, and deletions) to its sequence. Unlike *de-novo* design, which seeks to generate entirely new sequences, protein optimization starts from a known functional protein that already satisfies certain requirements, such as binding to a target or exhibiting measurable activity. The goal is then to explore its local neighborhood by applying mutations that preserve overall structure and function while improving targeted properties (Listov et al., 2024).

In recent years, protein language models (PLMs) have emerged as successful tools for protein optimization (Hie et al., 2024; Lin et al., 2023; Madani et al., 2023). While PLMs capture rich functional and evolutionary information, existing models are not naturally aligned with the local-mutation setting under which protein optimization operates. Autoregressive models regenerate an entire protein one residue at a time, making it difficult to control how many mutations are introduced and therefore how far the generated sequence drifts from the starting protein. This is particularly limiting in optimization settings, where the goal is often to introduce only a few local mutations. Masked language models are better suited for local editing, but require substitution sites to be specified externally and do not naturally support insertions, deletions, or variable-length generation.

---

[*]Equal contribution.

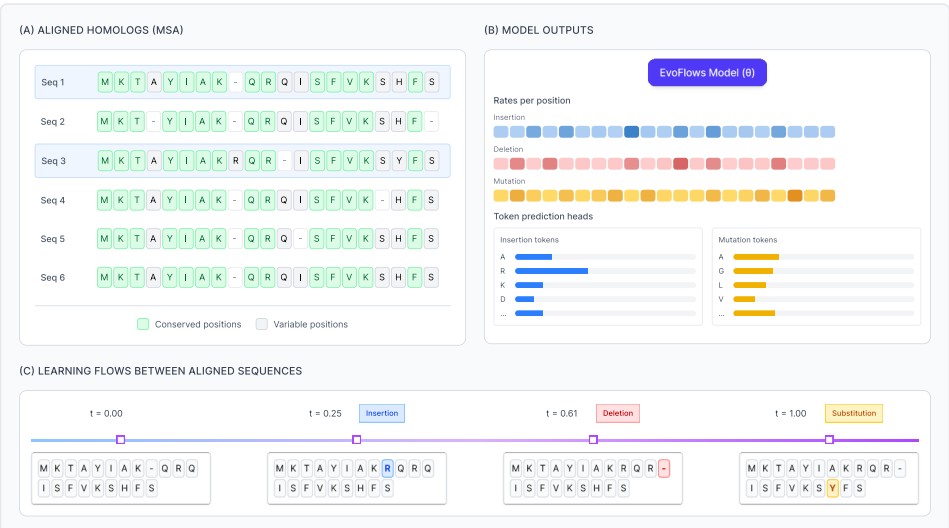

Figure 1: *Overview of EvoFlows.* Edit process on two sequences from a set of homologs.

Given the outsized impact of optimization in pre-clinical drug development pipelines (Paul et al., 2010), designing fit-for-purpose models is highly desirable. Before being conditioned on a specific task (Gruver et al., 2023; Widatalla et al., 2024), such a model should first generate function-preserving variants of a protein for unguided exploration (Hie et al., 2024) and serve as a strong foundation for fine-tuning. Protein optimization would benefit from models that combine the local-editing capabilities of masked language models with the variable-length generation abilities of autoregressive models. Such models should support local editing without requiring externally specified mutation locations and should naturally handle insertions and deletions. The latter is particularly important for protein families such as antibodies, where insertions and deletions are common (Rock et al., 1994).

To this end, we introduce EvoFlows, a discrete flow-matching approach (Gat et al., 2024; Havasi et al., 2025) for protein optimization that learns mutation-based sequence-to-sequence transition rules. EvoFlows generates protein variants by applying mutations (including insertions and deletions) to an existing protein (see Figure 1), rather than modeling token-level likelihoods or denoising corrupted inputs. We show that EvoFlows effectively captures evolutionary patterns and yields high-quality generative models that natively support variable-length protein sequences while matching the performance of the current state-of-the-art PLMs.

We demonstrate the following results:

- Edit flows, trained in a sequence-to-sequence setup, reliably recovers target edit distributions, which we confirm in a controlled deterministic setting with known ground-truth edits (Section 4.1).

- EvoFlows, trained on natural protein homologs, captures family-level sequence distributions competitive with state-of-the-art baselines, while generating more diverse variants without requiring externally specified mutation sites (Section 4.2).

- We propose a new grid-free inference procedure better suited for vectorized operations.

## 2 RELATED WORK

## 2.1 Protein Language Models (PLMs)

Protein language models (PLMs) learn representations of protein sequences by pre-training on large corpora of unlabeled sequences. Common pre-training methods include masked language modeling (MLM) (Brandes et al., 2022; Elnaggar et al., 2022; Verkuil et al., 2022), autoregressive PLMs (Chen et al., 2024; Ferruz et al., 2022; Jr and Bepler, 2025; Madani et al., 2023), and discrete diffusion-based sequence models (Alamdari et al., 2024; Hallee et al., 2025). These models have been shown to capture structural, functional, and evolutionary information, and serve as foundations for downstream prediction and generation tasks.

**Autoregressive PLMs** parameterize a distribution over protein sequences as

$$p_\theta(x) = \prod_{i=1}^{L} p_\theta(x_i \mid x_{<i}),$$
(1)

where $x = (x_1, ..., x_L) \in \mathcal{A}^L$ is a protein sequence over the amino acid alphabet $\mathcal{A}$, and each conditional probability $p_\theta(x_i \mid x_{<i})$ is a categorical distribution over amino acids. We denote $x_{<i}$ as a sequence where positions $\geq i$ are masked.

Autoregressive models have been successful for *de novo*-like tasks such as enzyme generation with weak conditioning (Madani et al., 2023; Munsamy et al., 2024). However, they generate entire sequences, making it difficult to control where mutations occur and how many mutations are introduced, making them less suitable for protein optimization tasks that only require a few targeted mutations. Additionally, methods generating local variants rely on conditioning on one (Chen et al., 2024) or multiple context sequences (Jr and Bepler, 2025) but require the model to generate the entire sequence to obtain a few mutations, which is inefficient.

**Masked PLMs** is often perceived as a more parsimonious approach for protein variant generation because it directly edits a subset of positions while preserving the rest of the sequence. MLMs instead learn the conditional distributions of the form

$$p_\theta\big(x_i | x_{\setminus i}\big), \quad i = 1, ..., L,$$
(2)

where $x_{\setminus i}$ denotes the sequence with position $i$ masked. As in the autoregressive case, the model outputs a categorical distribution over amino acids at each position.

Masked PLMs define residue-level conditional distributions rather than a distribution over protein sequences. While they can be used for sequence generation through iterative decoding, this requires specifying mask locations and assumes a fixed sequence length.

**Discrete diffusion models** generalize masked and autoregressive language modeling by introducing an explicit generation path through a sequence of intermediate states. They define a stochastic forward corruption process over discrete sequences and learn to invert this process through denoising (Austin et al., 2021; Hoogeboom et al., 2022).

Formally, a diffusion model defines a forward Markov process that progressively corrupts a clean sequence $x^{(0)}$ and learns the reverse process

$$p_\theta\big(x^{(t-1)} | x^{(t)}\big),$$
(3)

which iteratively denoises toward the data distribution.

In protein diffusion models, the corruption is often defined by iteratively introducing "absorbing" states ([MASK] tokens)(Alamdari et al., 2024) to which valid tokens are converted. This framework is a generalization that encompasses both autoregressive (left-to-right denoising paths) and MLM (single step in order-agnostic paths).

This generalization also appears in potential applications of diffusion models in protein optimization: conditioning on a starting sequence requires applying some level of corruption, which raises the same issue for picking *where* that corruption should be applied like for masked PLMs. On the other hand, starting from a fully corrupted state requires the same level of unnecessary computation and conditioning as autoregressive models.

## 2.2 EVOTUNING

Evotuning (Alley et al., 2019) adapts pre-trained PLMs to a specific protein family by further training on sequences drawn from a homologous sequence set, such as those retrieved from a protein database by sequence-similarity search (Steinegger and Söding, 2017), using the same self-supervised learning objective. Evotuned models have been shown to better capture family-specific constraints and improve performance on downstream prediction and design tasks (Alley et al., 2019). Given the opportunity for improvement on tasks limited to a single protein family, foundational PLMs should be evotuned when compared to family-specific models like EvoFlows.

## 2.3 DISCRETE FLOW MATCHING (DFM) AND EDIT FLOWS

Discrete Flow Matching (Gat et al., 2024) defines a continuous-time Markov chain (CTMC) $X_t$ over sequences of tokens in a discrete alphabet $\mathcal{A}$. DFM models approximate a joint distribution over pairs of sequences $\pi(x, x')$ by learning a local flow $u_t$ such that

$$\mathbb{P}(X_{t+h} = x_{t+h} | X_t = x_t) = \delta[x_{t+h} - x_t] + h u_t(x_{t+h} | x_t) + o(h),$$
$$\mathbb{P}(X_0 = x, X_1 = x') = \pi(x, x'). \tag{4}$$

This permits the transport of source samples $x_0 \sim p_0(X_0)$ to target samples $x_1 \sim p_1(X_1)$ such that pairs follow the joint distribution $\pi(x_0, x_1)$.

While the original formulation of DFM was limited to fixed-length sequences, Havasi et al. (2025) introduce edit flows, a variant where transitions encoded by $u_t$ are extended to elementary edit operations (substitutions, insertions, deletions).

This target rate is generated by defining an extended space $\mathcal{Z} = \mathcal{A} \cup \{\varepsilon\}$ where $\varepsilon$ represents an empty character that enables token-wise alignment of sequence pairs $(x_0, x_1)$ by lifting them to $z_0, z_1 \in \mathcal{Z}^*$, two sequences of equal length. A conditional probability is then defined from a continuous, bijective, increasing schedule $\kappa_t : [0, 1] \to [0, 1]$

$$p_t(z | z_0, z_1) = \prod_{i \in [\![1, \dots, |z|]\!]} \left( (1 - \kappa_t) \, \delta_{z_0^{(i)}} \big( z^{(i)} \big) + \kappa_t \, \delta_{z_1^{(i)}} \big( z^{(i)} \big) \right). \tag{5}$$

This leads to a conditional rate through differentiation and can be marginalized to obtain $u_t$.

Importantly, because edits are always elementary, $u_t$ can be expressed in terms of sequences in $\mathcal{A}^*$ exclusively. Given a distribution $\pi(z_0, z_1)$ that generates aligned pairs, the model $u_t^\theta$ is trained by optimizing a Bregman divergence for all times $t$:

$$\mathcal{L} = \mathop{\mathbb{E}}_{\substack{\pi(z_0, z_1) \\ t \sim \mathcal{U}(0,1) \\ p_t(z | z_0, z_1)}} \left( \sum_{x' \neq x} u_t^\theta(x' | x) - \frac{\dot{\kappa}_t}{1 - \kappa_t} \sum_{\substack{i \in [\![1, \dots, |z|]\!] \\ z^{(i)} \neq z_1^{(i)}}} \log u_t^\theta \Big( x \big( z, i, z_1^{(i)} \big) | x(z) \Big) \right), \tag{6}$$

where $x(z)$ is the $\mathcal{A}^*$-sequence obtained from $z$ by removing $\varepsilon$ tokens and $x(z, i, c)$ is the $\mathcal{A}^*$-sequence obtained by inserting $c$ at position $i$ before removing $\varepsilon$. Note that the first sum is tractable because $u$ is non-zero only for pairs of sequences that differ by an elementary edit.

We find that Edit Flows offer a combination of features that existing PLMs do not cover: unlike MLMs, they support variable-length sequence generation and do not rely on *ad hoc* mask insertions and, unlike autoregressive models they modify sequences locally. In particular, the progressive addition of elementary edits mimics the way single-amino-acid mutations connect sequences through evolution or protein engineering. We therefore propose that PLMs based on Edit Flows can fill the functionality gap we observe in the protein optimization space.

## 3 EVOFLOWS

We introduce EvoFlows, protein language models designed for protein optimization. In a nutshell, EvoFlows are discrete edit-flow models (Havasi et al., 2025) that connect evolutionarily-related sequences, homologs, to each other through a series of mutation operations: substitutions, deletions and insertions. As described in Section 2.3, we need to specify a data distribution $\pi$ over $\mathcal{Z}^* \times \mathcal{Z}^*$, which we chose to do by first specifying a distribution over $\mathcal{A}^* \times \mathcal{A}^*$ and an alignment procedure that inserts $\varepsilon$ tokens in protein sequences.

### 3.1 HOMOLOGOUS PROTEIN PAIR DISTRIBUTIONS

An EvoFlow model should enable sampling variants of a template protein sequence $x$ that are likely to preserve its function. In the language of Section 2.3, this means that we should define $\pi(x, x')$ to capture some notion of $\mathbb{P}(x$ has a similar function to $x')$. Protein homology (meaning that two or more proteins share a common ancestor through evolution) provides a useful approximation for biochemical function similarity: related species produce similar proteins which perform analogous tasks in their respective organisms. These evolutionary relationships are established through patterns of similarity between protein sequences across species, where functional components of a protein tend to be more strongly conserved and mutations are biased toward chemically similar amino-acids (Henikoff and Henikoff, 1992).

We therefore propose the following procedure to generate the pairwise distribution $\pi(x, x')$ over pairs of protein sequences: given a starting protein $x^\dagger$, we use protein homolog search tools such as mmseqs2 (Steinegger and Söding, 2017) to query large protein databases such as UNIREF30 (Suzek et al., 2007) and OAS (Olsen et al., 2022) for likely-related natural sequences. This cluster of similar protein sequences $R(x^\dagger)$ is taken as an empirical distribution over proteins that have an evolutionary relationship to $x^\dagger$:

$$p_{x^\dagger}(x) = \frac{1}{|R(x^\dagger)|} \sum_{x' \in R(x^\dagger)} \delta_{x'}(x). \tag{7}$$

So that our models capture sequence-to-sequence relationships characteristic of protein homology, we define the target $\pi(x, x') \propto \mathbb{1}(x$ homologous to $x')$, which we approximate as

$$\pi_{x^\dagger}(x, x') = p_{x^\dagger}(x)p_{x^\dagger}(x'). \tag{8}$$

This leads to EvoFlows learning to sample homologs $x'$ of a starting sequence $x$.

### 3.2 PAIRWISE PROTEIN ALIGNMENT

The target path distribution is defined as an expectation over a distribution of aligned pairs $\pi(z_0, z_1)$, where $z_0$ and $z_1$ represent source and target sequences augmented with an empty token $\varepsilon$ to ensure that they have equal length. There is a natural notion of pairwise alignment for proteins, which is part of how sequence homology is estimated (Needleman

and Wunsch, 1970; Smith and Waterman, 1981). This means that we can define sequence-to-sequence edit flow models in the protein space that rely on standard protein sequence alignment tools, where elementary edits including insertions and deletions correspond to actual biochemical events (Henikoff and Henikoff, 1992). We use the Needleman-Wunsch algorithm to compute alignments between pairs of protein sequences to produce augmented sequences in $\mathcal{Z}$.

## 3.3 Inference sampling

At inference time, our generative model draws samples from $p_1$ by sampling a realization of the corresponding CTMC $x_t$ and returns $x_1 \sim p_1$:

- Initialize: Sample $x_0 \sim p_0$ and set $t_0 = 0$.
- Time-to-event: At time $t_n$, sample the duration $\Delta_n$ until the next event. Let $\tau$ denote the first-event time after $t_n$. The CDF for $\tau$ is

$$\mathbb{P}\big(\tau \leq T \mid x_{t_n}\big) = 1 - \exp\left(\int_{t_n}^{t_n+T} u_s\big(x_{t_n}|x_{t_n}\big)\,\mathrm{d}s\right). \tag{9}$$

  Intuitively, higher total rate $u_s\big(x_{t_n}|x_{t_n}\big)$ makes the next mutation happen sooner. We sample $\Delta_n$ via inverse CDF: draw $U \sim \mathrm{Uniform}(0,1)$ and numerically integrate until

$$\log(U) - \int_{t_n}^{t_n+\Delta_n}\big(u_s\big(x_{t_n}|x_{t_n}\big)\big)\,\mathrm{d}s = 0. \tag{10}$$

  This lets us sample the next event time directly from the total rate predicted by the model.

- Denote $t_{n+1} = t_n + \Delta_n$. If $t_{n+1} < 1$, sample

$$x_{t_{n+1}} \sim -\frac{u_{t_{n+1}}\big(\cdot\,|x_{t_n}\big)}{u_{t_{n+1}}\big(x_{t_n}|x_{t_n}\big)}, \tag{11}$$

  and continue. This step selects which specific mutation is applied next. Otherwise, return $x_{t_n}$ as $X_1$.

Note that this method is formally very close to the Euler integration proposed by Havasi et al. (2025) but replaces repeated Bernoulli variable sampling with a single uniform sampling. This approach has the practical benefit that it does not require adding potentially large sub-leading terms to (4), otherwise needed to avoid pathological distributions.

We additionally introduce a clock normalization hyperparameter that applies a length-normalized rate scaling, allowing control over the number of mutations the model performs independently of sequence length.

## 3.4 Architecture

EvoFlows parameterizes the rate function $u_t^\theta(x' \mid x)$ using an architecture that operates directly on sequences $x \in \mathcal{A}^*$ and continuous time $t \in [0,1]$. We use the encoder trunk of a pre-trained ESM-2 model (Lin et al., 2023) to compute a time-independent sequence embedding which is then made time-dependent using Feature-wise Linear Modulation (FiLM) (Perez et al., 2018). We finally use shallow MLPs as prediction heads to map token

Table 1: *Mutation classification on the deterministic benchmark.* Precision, recall, and F1-score for each class (substitution, insertion, deletion, and no-op), along with ground-truth class prevalence. Values are reported with bootstrap confidence intervals. Performance is highest for no-ops, while insertion, substitution, and deletion remain well separated, indicating reliable discrimination between mutation types in the deterministic setting.

| Mutation Type | Precision | Recall | F1-Score | Prevalence |
|---|---|---|---|---|
| No-op | $0.982 \pm 0.001$ | $0.983 \pm 0.000$ | $0.982 \pm 0.000$ | 0.888 |
| Insertion | $0.820 \pm 0.005$ | $0.796 \pm 0.006$ | $0.808 \pm 0.005$ | 0.047 |
| Substitution | $0.804 \pm 0.005$ | $0.843 \pm 0.007$ | $0.823 \pm 0.005$ | 0.043 |
| Deletion | $0.910 \pm 0.006$ | $0.850 \pm 0.007$ | $0.879 \pm 0.006$ | 0.022 |

embeddings to rates for individual mutations. More details on the parametrization can be found in Section A

Importantly for practical applications, the late application of the FiLM time embedding means that the cost of updating the time $t$ for a fixed input is negligible. This makes inference as described in Section 3.3 efficient despite the numerical integral potentially requiring many forward passes.

# 4 RESULTS

Unlike Havasi et al. (2025), where source distributions correspond to noise, both marginals in our flow describe natural protein sequences. We first validate edit flows on a synthetic task with known expected mappings, then evaluate EvoFlows on six protein homolog datasets.

## 4.1 DETERMINISTIC SETUP

We construct a synthetic dataset with deterministic mutation patterns to evaluate the model in a controlled setting. From a natural protein sequence $z_0$, we generate $z_1$ through position-dependent rules:

- If $z_0[i] = A$, include $\mathrm{Sub}(i+5, H)$.
- If $z_0[i] = C$, include $\mathrm{Ins}(i-2, S)$.
- If $z_0[i] = G$ and $z_0[j] = L$ and $z_0[k] = K$ for some $j < i < k$, include $\mathrm{Del}(i)$.

Mutations are applied in order: insertions (increasing index), deletions, then substitutions. This yields a unique $z_1$ for each $z_0$, enabling precise assessment of mutation type, position, and amino acid identity, unlike natural MSAs where alignments may admit multiple explanations. Results across sequence lengths and clock normalization factors are shown in Figure 2, Figure 4, and Table 1.

## 4.2 EVOFLOWS

We evaluate EvoFlows on 6 seed proteins: enzymes, growth factors, and antibody fragments (VHH and ScFv). Each homolog set is split into train, inference, and holdout. From each inference sequence $x_0$, we generate $x_1$ using 6 methods: Random inference homolog pairing, EvoFlows, EvoDiff-MSA (Alamdari et al., 2024), Evotuning, Evotuning with forced substitutions, and random mutations. We compare the resulting $\{x_1\}$ to the holdout split as a proxy target distribution, matching the expected number of mutations per sequence across

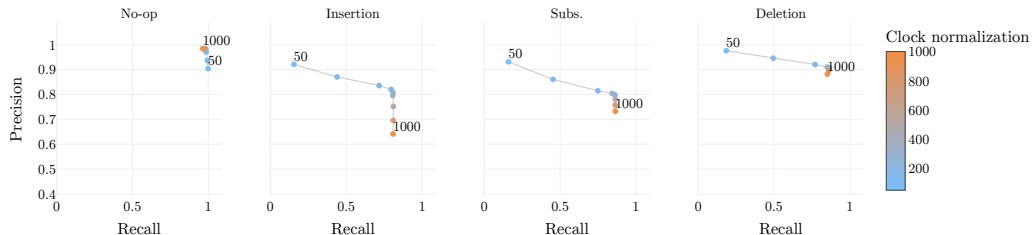

Figure 2: *Precision-recall trade-offs under clock normalization.* Precision-recall curves for each mutation type and no-ops on the deterministic benchmark, which shows how clock normalization controls the trade-off between recall and precision. While no-op predictions remain stable, insertion, substitution, and deletion exhibit systematic precision-recall shifts as the clock normalization varies, highlighting its role in calibrating mutation selection behavior.

methods (except random pairing). Random inference homolog pairing pairs each inference sequence with a randomly selected homolog from the same protein family. This provides an approximate upper bound on family-level similarity, since both sequences are drawn from the natural homolog distribution. Random mutations instead serve as a lower bound that does not preserve family-level structure.

EVOFLOWS INFERENCE

We sample $x_1 \sim p_\theta(\cdot \mid x_0)$ from a trained EvoFlows model.

**Data and pair construction.** Homologs are obtained via iterative profile search against UniRef30 with profile expansion and realignment, filtering for query coverage $\geq 0.8$ and E-value $\leq 10^{-1}$. We additionally search ColabFold's environmental database to increase evolutionary depth. Training pairs are formed by enumerating all unordered sequence pairs; each pair $(x_0, x_1)$ is globally aligned via Needleman-Wunsch to yield $(z_0, z_1) \in \mathcal{A}_\varepsilon^L \times \mathcal{A}_\varepsilon^L$. Dataset details are provided in Table 2.

4.3 BASELINES

**Random inference homolog pairing.** Pairs each inference sequence with a randomly selected homolog from the same protein family. This does not generate new variants, but instead provides a reference for the level of similarity expected between naturally occurring homologs.

**Evotuned PLM.** We evotune ESM-2-650M on the same homologs used for EvoFlows training. Substitution positions are sampled from a positional distribution derived from per-column entropy:

$$H(\ell) = - \sum_{\{a \in \mathcal{A}\}} p_\ell(a), \log(p_\ell(a) + \varepsilon), \tag{12}$$

where $p_\ell(a)$ is the amino acid frequency in aligned column $\ell$ (excluding gaps). Entropy weights are normalized and mapped to ungapped coordinates. Masked positions are iteratively infilled using the evotuned MLM with temperature scaling.

**Evotuned PLM with forced substitutions.** Forces substitutions at masked positions by blocking infilling of the original amino acid.

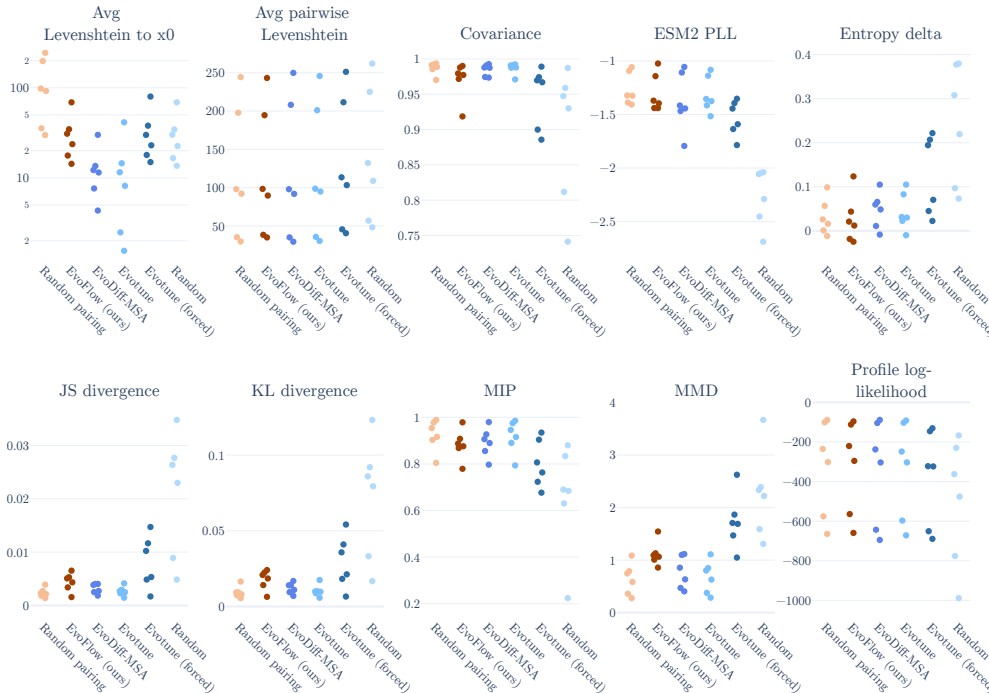

Figure 3: Comparison of EvoFlows and baseline methods across evaluation metrics. Each point corresponds to one homolog dataset derived from one of the seed proteins described in Section C. EvoFlows generates variants that remain close to the holdout distribution while exploring a larger local neighborhood around the starting protein. EvoDiff-MSA achieves competitive distributional quality on several metrics, but typically explores a smaller neighborhood around the starting protein and produces less diverse sets of variants than EvoFlows. In practice, EvoDiff-MSA also struggles to introduce insertions: adding masked positions inside runs of gaps often results in those positions being regenerated as gaps rather than amino acids, and masking entire runs of gaps makes it difficult to control the number of insertions. Evotuning without forced substitutions also achieves competitive scores, but introduces very few mutations, in some cases as low as 1.5 mutations per sequence on average. Forcing substitutions increases mutation diversity but substantially degrades distributional quality. Random mutations perform worst across all metrics. See Figure 5 for per-dataset breakdowns.

**EvoDiff-MSA.** Uses the order-agnostic discrete diffusion model trained on MSA subsets (OA-DM-MAXSUB) (Alamdari et al., 2024). To generate variants, we build an MSA from the same homologs used for EvoFlows training and evotuning, then jointly align the MSA with $x_0$ using MAFFT. Mutation positions are selected using the same entropy-weighted positional distribution as the Evotuned PLM, but applied directly in aligned coordinates. That is, gapped positions can be masked, enabling insertions, and gapped tokens can be infilled, enabling deletions. Masked positions are infilled by EvoDiff-MSA conditioned on up to 64 randomly subsampled MSA sequences.

**Random mutations.** Applies the same expected mutation count, but samples mutation positions, mutation types (substitution, insertion, deletion), and replacement amino acids uniformly.

**Benchmarking indels.** EvoFlows natively supports substitutions, insertions, and deletions. By contrast, the MLM baselines only support substitutions, since no widely used

method exists for generating indels with MLMs. As a result, comparisons involving MLM baselines are limited to substitutions.

### 4.4 Inference runtime

EvoFlows requires a number of forward passes proportional to the expected number of edits. Autoregressive models require one pass per token in the full sequence resulting in a number of passes proportional to sequence length, and EvoDiff-MSA must repeatedly mask and regenerate all candidate edit positions, including entire runs of gaps to get insertions with substantially weaker control over the final edit count.

## 5 Conclusion

We introduced EvoFlows, an edit-based discrete flow matching framework that learns protein transition rules via substitutions, insertions, and deletions. To our knowledge, EvoFlows is the first method to directly model insertions and deletions as explicit edit operations in a template-based protein optimization setting. Supporting insertions and deletions in a template-based setting typically requires ad hoc masking schemes and offers limited control over edit placement and edit count. Masked language models are limited to substitutions at externally specified positions, autoregressive models require full sequence regeneration and offer no natural mechanism to control the number of mutations introduced, and discrete diffusion models inherit equivalent limitations when conditioned on a template.

We extend the original edit flow framework in two key ways. First, protein sequences are a particularly natural fit for the edit flow framework: unlike text or code, protein evolution operates through discrete, well-defined mutations, and large datasets of aligned protein families are readily available for training. Second, we introduce a deterministic benchmark that for the first time validates that edit flows can reliably learn local transition distributions in a seq2seq setting, establishing the correctness of the framework before applying it to the setting of protein homologs. Across six diverse protein families spanning enzymes, growth factors, and antibody fragments, EvoFlows generates variants that are non-trivial and natural-like, exploring significantly larger mutational neighborhoods than state-of-the-art baselines while remaining close to the holdout distribution.

Our evaluation is currently limited to *in silico* analysis. While the generated variants exhibit natural-like properties, wet-lab validation remains essential to confirm functional fitness.

EvoFlows itself learns a prior over plausible local sequence changes rather than directly optimizing for particular engineering objectives. Looking forward, we see train-time and inference-time conditioning as promising directions for practical protein engineering: property-based guidance, sequence constraints (motifs, mutation budgets), and metadata conditioning (species, gene ontology). Such mechanisms would enable EvoFlows to serve as a foundation for guided protein optimization, where the goal is to generate local variants of a known template sequence.

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

## A   ARCHITECTURE DETAILS

**Sequence embedding:** Given a sequence $x_t \in \mathcal{A}^L$, we compute token embeddings using a pretrained ESM-2 model,

$$h_t = \text{ESM2}(x_t) \in \mathbb{R}^{L \times D}, \tag{13}$$

where $D$ is the dimension of the embeddings. To maintain consistency with ESM-2, the embedder includes explicit start and end tokens. During training, the ESM-2 encoder weights are finetuned jointly with the EvoFlows model rather than frozen.

**Time conditioning:** We embed the time $t$ using a sinusoidal embedding followed by an MLP:

$$\tau_t = \text{MLP}(\text{Sinusoidal}(t)) \in \mathbb{R}^D. \tag{14}$$

This time embedding is used to condition the token representations via Feature-wise Linear Modulation (FiLM) (Perez et al., 2018), which learns a feature-wise affine transformation that maps $\tau_t$ to scale and shift vectors $(\gamma_t, \beta_t) \in \mathbb{R}^D$. The token embeddings are transformed as

$$\tilde{h}_t[i] = h_t[i] \odot (1 + \gamma_t) + \beta_t, \tag{15}$$

and are shared across all heads.

Notably, the proposed time-conditioning mechanism enables efficient changing of time for fixed inputs, which is particularly useful for training with multiple time samples per sequence in a batch.

**Rate prediction:** The rate $u_t^\theta(x'|x)$ is obtained through a similar parametrization as what is adopted by Havasi et al. (2025):

$$u_t^\theta(x'|x) = \begin{cases} \lambda_t^{\text{sub}}(\tilde{h}_t[i]) Q_t^{\text{sub}}(\tilde{h}_t[i], i, a) \text{ if } x' = \text{sub}(x, i, a) \\ \lambda_t^{\text{ins}}(\tilde{h}_t[i]) Q_t^{\text{ins}}(\tilde{h}_t[i], i, a) \text{ if } x' = \text{ins}(x, i, a) \\ \lambda_t^{\text{del}}(\tilde{h}_t[i]) Q_t^{\text{del}} \text{ if } x' = \text{del}(x, i) \\ 0 \text{ otherwise} \end{cases}, \tag{16}$$

where $\text{sub}(x, i, a)$ denotes $x$ transformed by a substitution by the token $a$ at position $i$, $\text{sub}(x, i, a)$ denotes $x$ transformed by an insertion of the token $a$ after position $i$ and $\text{del}(x, i)$ denotes $x$ transformed by a deletion at position $i$.

Both $\lambda_t^\bullet$ and $Q_t^\bullet$ are defined as shallow MLPs and denote respectively positional rates and distributions over tokens. The positional rates $\lambda_t^\bullet$ are normalized using either a softplus transformation or a bounded sigmoid mapping to ensure the rates are positive and numerically stable. The token distributions $Q_t^\bullet$ are parametrized as ESM2 language prediction heads.

## B    Evaluation Metrics

In this section, we describe the selected metrics for sequence and dataset distribution evaluation and comparison in detail. Generative model evaluation has been notoriously difficult, and we aim to choose metrics and distances suitable for the proposed task of protein sequence generation.

### B.1    Dataset Heuristics

We hypothesize that a well performing, unconditioned generation should be distributionally indistinguishable from a subsampled hold-out set of the multiple sequence alignment used for training. The generative process should thus recreate this set of defined heuristics. At a sequence level, we consider the distribution of sequence lengths, the overall distribution of amino acid residues, and the per-position distribution of amino acids and gaps with respect to a fixed sequence alignment.

### B.2    Model-Based Pseudo Log Likelihood

It is possible to quantify the likelihood of a sequence under a probabilistic model. The most commonly known method related to LLMs is perplexity (PPL, applicable to autoregressive models), or pseudo log likelihood (PLL, applicable to masked language models). For both

models the cumulative likelihood of a generative path is calculated. While AR models can be traversed in a single, well-defined path, this does not hold true for MLM. For the latter, we choose a single random order over individually masked positions to calculate the pseudo log likelihood.

### B.3 Covariance and Mutual Information

A strong indicator for models correctly capturing signals connected to coevolution is the proper covariance of individual positions in an alignment. Covariance can be an indicator of sequences that co-evolved with epistatic changes, for instance to maintain structural contacts.

In natural protein evolution, amino acid substitutions do not occur independently. When a mutation changes one position in a protein structure, compensatory mutations at other positions often follow to maintain protein stability and function. These co-evolving positions exhibit statistical dependencies that can be detected as covariance in multiple sequence alignments. A generative model that accurately captures these evolutionary constraints should reproduce the covariance patterns observed in natural protein families.

**4D Covariance.** Given a set of one-hot-encoded sequences $\mathcal{X} = \{x_1, ..., x_N\} \subset \{0,1\}^{L \times 21}$ from a joint alignment, we define the positional amino acid frequency $f_i(a)$, the joint frequency $f_{ij}(a,b)$, and the covariance $C_{ij}(a,b)$ as:

$$f_i(a) = \frac{1}{N} \sum_{n}^{N} x_{nia} \in \mathbb{R}^{L \times 21},$$

$$f_{ij}(a,b) = \frac{1}{N} \sum_{n}^{N} x_{nia} x_{njb} \in \mathbb{R}^{L \times L \times 21 \times 21}, \tag{17}$$

$$C_{ij}(a,b) = f_{ij}(a,b) - f_i(a) f_j(b) \in \mathbb{R}^{L \times L \times 21 \times 21}.$$

**Positional Interaction Strength.** The 4-dimensional tensor $C_{ij}(a,b)$ captures the per-position, per-amino-acid covariance over a dataset. We can contract this information by using the Frobenius norm over the last two dimensions, which can be interpreted as contracting the per-residue coupling into an overall interaction strength at a given pair of positions $i$ and $j$:

$$\|C_{ij}\|_F = \sqrt{\sum_{a,b} C_{ij}(a,b)^2}. \tag{18}$$

**Mutual Information with Average Product Correction.** An interpretable 2D representation of coevolution can also be constructed from information theory, as described by (Dunn et al., 2007). Using the definitions from equation (17), we can describe the mutual information $I_{ij}$ as:

$$I_{ij} = \sum_{a,b} f_{ij}(a,b) \log \frac{f_{ij}(a,b)}{f_i(a) f_j(b)}. \tag{19}$$

We denote column, row, and global means as

$$I_i = \frac{1}{L} \sum_{k=1}^{L} I_{ik}, \quad I_j = \frac{1}{L} \sum_{k=1}^{L} I_{kj}, \quad I = \frac{1}{L^2} \sum_{i,j} I_{ij}, \tag{20}$$

and define the Average Product Correction $\text{APC}_{ij}$ as

$$\text{APC}_{ij} = \frac{I_i I_j}{I}. \tag{21}$$

This lets us define MIp, the mutual information with average product correction, as

$$\text{MIp}_{ij} = I_{ij} - \text{APC}_{ij}. \tag{22}$$

## B.4   BLOSUM-CORRECTED KL DIVERGENCE

While the metrics described above characterize properties of individual datasets, this section focuses on distance measures that quantify the similarity between two datasets. These measures are essential for comparing synthetic sequences generated by our models against held-out natural sequences from the training MSA. The Kullback-Leibler (KL) divergence provides a principled information-theoretic approach to measure how one probability distribution differs from another.

Given two discrete probability distributions $p$ and $q$ over amino acid frequencies, the KL divergence is defined as:

$$D_{\text{KL}}(p \parallel q) = \sum_a p(a) \log \frac{p(a)}{q(a)}, \tag{23}$$

where the sum is taken over all amino acid types. The KL divergence is always non-negative and equals zero if and only if $p$ and $q$ are identical distributions. However, $D_{\text{KL}}$ is asymmetric: $D_{\text{KL}}(p \parallel q) \neq D_{\text{KL}}(q \parallel p)$ in general.

**The Zero-Frequency Problem and Biological Priors.** A critical challenge in computing KL divergence for protein sequences arises from the zero-frequency problem. In small datasets or at highly conserved positions, certain amino acids may not appear at all, leading to zero probabilities. When $q(a) = 0$ and $p(a) > 0$, the KL divergence becomes infinite, making comparisons impossible.

Additive smoothing (also known as Lidstone smoothing) addresses this by adding pseudo-counts to observed frequencies. Rather than using uninformative uniform pseudo-counts, such as `1/vocabulary_size`, we employ `BLOSUM62` background frequencies (Altschul et al., 2009) as biologically informed priors. These frequencies reflect the natural abundance of amino acids in protein databases and have been empirically validated across diverse protein families. The smoothed probability for amino acid $i$ is calculated as:

$$p_{a,\alpha} = \frac{x_a + \alpha \mu_a}{N + \alpha \cdot d}. \tag{24}$$

where $x_a$ is the observed count of amino acid $a$, $N$ is the total number of observations, $d = 21$ is the number of possible amino acids (20 standard amino acids plus gap), $\mu_a$ is the `BLOSUM62` prior frequency for amino acid $a$, and $\alpha$ is the pseudo-count weight parameter. We calibrated the gap frequency $\mu_{\text{GAP}}$ empirically across all MSAs used in our evaluation, finding a value of 0.7% to match most common cases using a coverage-based proxy for pairwise alignment sparsity.

**Biological Justification.** The use of `BLOSUM62` frequencies as priors is biologically motivated. These frequencies were derived from the BLOCKS database of conserved protein regions and reflect amino acid propensities averaged over many protein families. By incorporating this knowledge, our smoothing method:

1. Ensures biologically plausible probability estimates even with limited data

2. Weights pseudo-counts according to amino acid abundance in natural proteins rather than treating all amino acids equally

3. Provides a Bayesian interpretation with Dirichlet priors informed by empirical protein evolution

4. Reduces the variance of frequency estimates while introducing minimal bias, as the smoothed probabilities converge to maximum likelihood estimates as dataset size increases

This approach is particularly valuable when comparing generated sequences to natural sequences, as it ensures that distance measurements remain finite and meaningful even when datasets differ in size or sampling depth, while respecting the underlying biochemical constraints of protein composition.

### B.5 Spectrum Kernel Maximum Mean Discrepancy

The Maximum Mean Discrepancy (MMD) provides a principled framework for comparing probability distributions by embedding them into a Reproducing Kernel Hilbert Space (RKHS). This approach presents a powerful, alignment-free metric for comparing generated sequences against reference distributions when combined with the Spectrum kernel for protein sequences.

**Theoretical Foundation.** The Maximum Mean Discrepancy (MMD) measures how different two probability distributions are by embedding them into a rich function space called a Reproducing Kernel Hilbert Space (RKHS). Intuitively, MMD finds the function that best distinguishes samples from the two distributions and measures how different the average function values are (Gretton et al., 2012).

Given samples $X = \{x_1, ..., x_n\}$ from a reference distribution and $Y = \{y_1, ..., y_m\}$ from a generated distribution, the squared MMD decomposes into three intuitive terms:

$$\texttt{MMD}^2 = \underbrace{\mathbb{E}[k(x, x')]}_{\text{similarity within } X} + \underbrace{\mathbb{E}[k(y, y')]}_{\text{similarity within } Y} - 2 \underbrace{\mathbb{E}[k(x, y)]}_{\text{similarity between } X \text{ and } Y}, \tag{25}$$

where $k(\cdot, \cdot)$ is a kernel function measuring sequence similarity. When sequences within each set are similar to each other (high first two terms) but the two sets differ from each other (low cross-term), the MMD is large, indicating distributional mismatch. Conversely, when generated sequences are indistinguishable from reference sequences, MMD approaches zero.

A key theoretical guarantee is that when using a *characteristic* kernel, MMD equals zero if and only if the two distributions are identical. This ensures that any systematic difference between generated and reference sequences will be detected, making MMD a principled choice for evaluating generative models.

In practice, the empirical MMD can be computed using either a biased (V-statistic) or unbiased (U-statistic) estimator, differing in whether diagonal terms of the kernel matrices are included. We use the biased estimator, which guarantees non-negative $\texttt{MMD}^2$ values—important when taking the square root—and exhibits lower variance than the unbiased alternative (Gretton et al., 2012).

**The Spectrum kernel**, introduced by Leslie et al. (2002), represents sequences as vectors of $k$-mer counts and computes similarity via their inner product. For protein sequences, this provides an alignment-free measure of compositional similarity.

Given a sequence $x$, we define the feature map $\Phi_k : \mathcal{X} \to \mathbb{R}^{|\mathcal{A}|^k}$ where $\mathcal{A}$ denotes the amino acid alphabet (typically 20 standard residues). Each coordinate of $\Phi_k(x)$ counts the occurrences of the corresponding $k$-mer:

$$\Phi_k(x)_a = |\{i : x[i : i + k] = a\}|. \tag{26}$$

This counts how many times the $k$-mer $a$ appears as a contiguous substring in sequence $x$, where $x[i : i + k]$ denotes the substring starting at position $i$ with length $k$.

The Spectrum kernel is then the inner product of these feature vectors:

$$k(x, y) = \langle \Phi_k(x), \Phi_k(y) \rangle = \sum_{a \in \mathcal{A}^k} \Phi_k(x)_a \cdot \Phi_k(y)_a. \tag{27}$$

This kernel has several attractive properties for protein sequence analysis:

1. It is efficient to compute using sparse representations, avoiding the need to explicitly enumerate all $|\mathcal{A}|^k$ possible $k$-mers.

2. It makes no assumptions about the data distribution, unlike learned embeddings.

3. It captures local sequence composition.

4. Its simplicity ensures robustness when evaluating artificial sequences (Kucera et al., 2022).

## C  PROTEIN TYPES AND DATASETS

We use the following seed proteins to construct homolog datasets via iterative profile search.

### Anti-SARS-CoV-2 VHH (Ty1)

Ty1 is an alpaca-derived single-domain antibody (nanobody) that targets the receptor-binding domain (RBD) of the SARS-CoV-2 spike protein, directly preventing ACE2 engagement and neutralizing viral infection (Hanke et al., 2020). The 12.8 kDa nanobody binds an epitope accessible in both the "up" and "down" RBD conformations, sterically hindering host receptor binding. Cryo-electron microscopy structures reveal that CDR1 and CDR3 loops mediate the primary contacts with the spike protein.

### Serine-Pyruvate Aminotransferase (SPAT)

Serine-pyruvate aminotransferase (EC 2.6.1.51) is a pyridoxal phosphate-dependent enzyme that catalyzes the reversible transamination between L-serine and pyruvate to produce 3-hydroxypyruvate and L-alanine (BRENDA Enzyme Database, 2025). In humans, the enzyme localizes to peroxisomes where it participates in glyoxylate detoxification; functional deficiency causes primary hyperoxaluria type 1, characterized by calcium oxalate accumulation. The enzyme plays a key role in gluconeogenesis from serine in the liver.

### Haloalkane Dehalogenase DhaA

DhaA is a haloalkane dehalogenase from *Rhodococcus rhodochrous* that catalyzes the hydrolytic cleavage of carbon-halogen bonds in halogenated compounds (Koudelakova et al., 2013). The enzyme employs a catalytic pentad (Asp106, His272, Glu130, Asn41, Trp107) to perform nucleophilic substitution, forming a covalent alkyl-enzyme intermediate that is subsequently hydrolyzed. DhaA and its engineered variants are of biotechnological interest for bioremediation of industrial pollutants such as 1,2,3-trichloropropane.

### Anti-HER2 scFv

Single-chain variable fragments (scFvs) targeting HER2 consist of the variable heavy (VH) and light (VL) chains of an anti-HER2 antibody connected by a flexible glycine-serine linker (Koçer et al., 2021). HER2 (human epidermal growth factor receptor 2) is overexpressed in 20–25% of breast cancers, making it a critical target for molecular diagnostics and therapy. Anti-HER2 scFvs are used in antibody-drug conjugates and as targeting moieties for nanoparticle delivery systems.

### Chicken FGF2

Table 2: Summary statistics of homolog datasets used for training and evaluation. Query and target coverage values represent the fraction of the query and target sequences aligned in each homolog pair.

| Dataset | Number of Homologs | Sequence Length | Query Coverage | Target Coverage |
|---|---|---|---|---|
| Anti-SARS-CoV-2 VHH | 3335 | $109.2 \pm 4.3$ | $0.88 \pm 0.01$ | $1.00 \pm 0.01$ |
| Anti-EphA2 VHH | 24304 | $118.2 \pm 3.7$ | $1.00 \pm 0.00$ | $1.00 \pm 0.00$ |
| Serine-Pyruvate Aminotransferase | 6565 | $363.0 \pm 16.7$ | $0.93 \pm 0.03$ | $0.91 \pm 0.10$ |
| Chicken FGF2 | 1435 | $141.6 \pm 11.4$ | $0.88 \pm 0.06$ | $0.71 \pm 0.19$ |
| Haloalkane Dehalogenase DhaA | 4697 | $282.2 \pm 11.5$ | $0.96 \pm 0.03$ | $0.91 \pm 0.10$ |
| Anti-HER2 scFv | 10979 | $246.4 \pm 4.8$ | $0.99 \pm 0.01$ | $1.00 \pm 0.00$ |

Fibroblast growth factor 2 (FGF2) from *Gallus gallus* is a multifunctional growth factor involved in cell proliferation, differentiation, angiogenesis, and tissue repair (UniProt Consortium, 2025). The protein signals through FGF receptors 1–4 and is essential for normal embryonic development. In chickens, three FGF2 isoforms (18.5, 20.0, and 21.5 kDa) are produced by alternative translation initiation during embryogenesis.

**Anti-EphA2 VHH**

Anti-EphA2 nanobodies are single-domain antibodies targeting ephrin type-A receptor 2, a receptor tyrosine kinase overexpressed in breast, prostate, lung, and bladder cancers (Roovers et al., 2011). Their small size ( 15 kDa), high stability, and superior tissue penetration make them well-suited for targeting solid tumors. These nanobodies have been developed for immunoliposomal drug delivery and as targeting agents for antibody-directed nanotherapeutics.

**Dataset Statistics** The number of homologs, sequence length distribution, and other relevant statistics for each seed protein dataset are detailed in Table 2.

C.1 SUPPLEMENTARY FIGURES

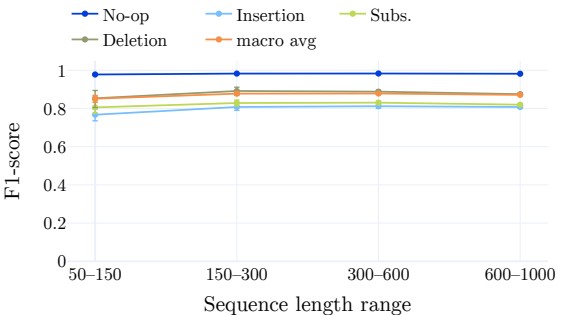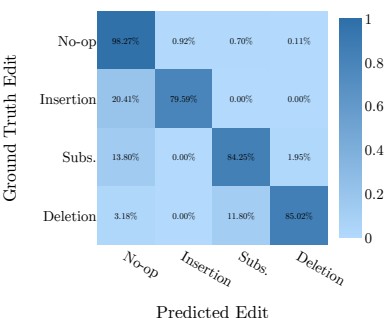

Figure 4: *Performance on the deterministic benchmark.*: *(Left)* F1-score for each mutation type as a function of sequence length. Performance remains stable across sequence lengths, indicating that mutation prediction accuracy does not degrade for longer sequences. *(Right)* Confusion matrix showing the distribution of predicted mutation types versus ground-truth mutations (no-op, insertion, substitution, deletion). The model correctly identifies most operations, with remaining errors primarily corresponding to confusions with the no-op class.

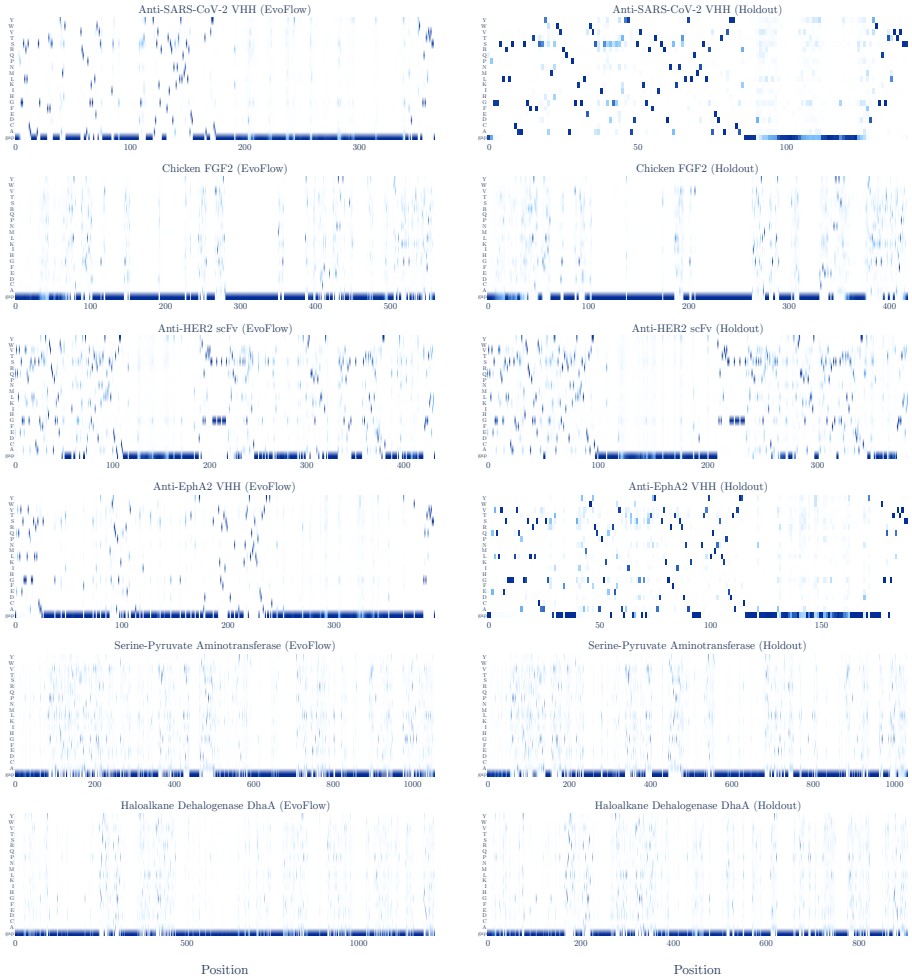

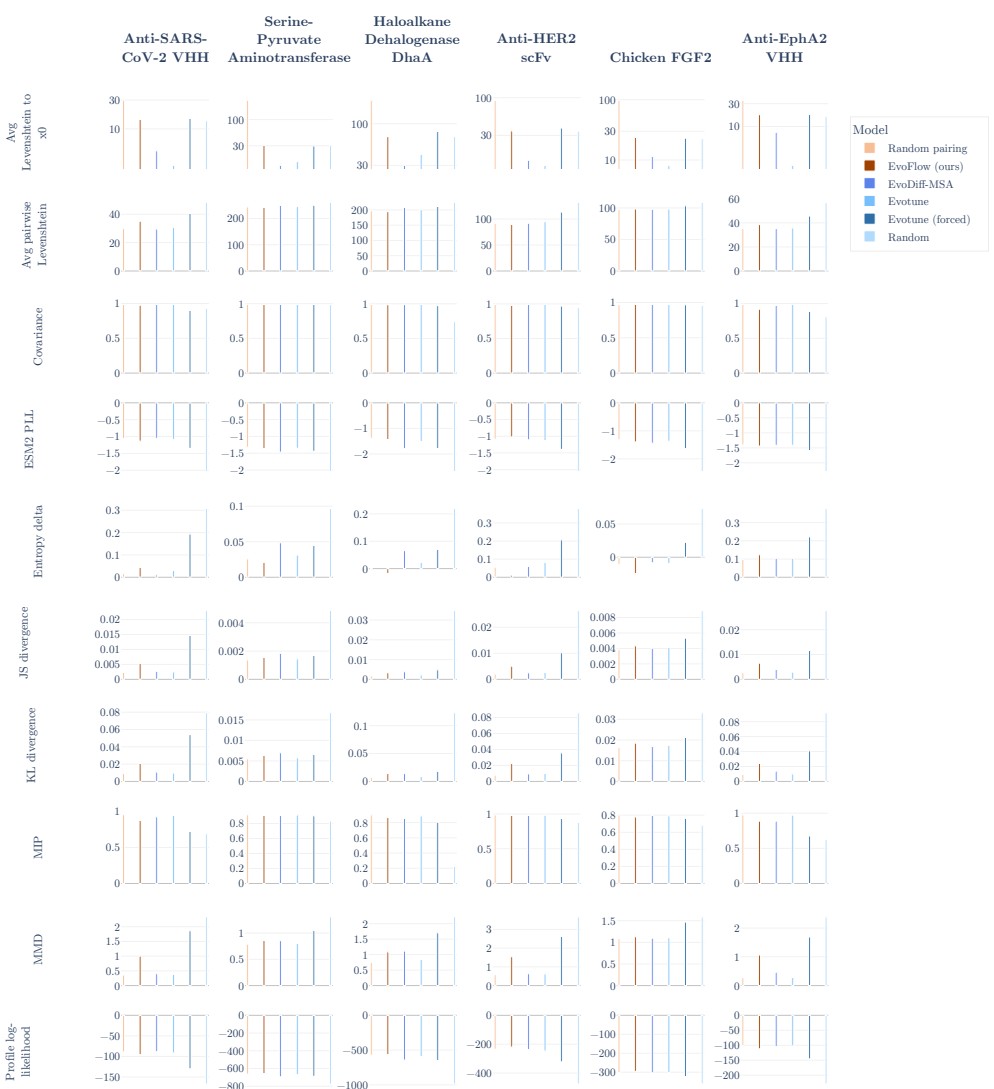

Figure 5: *Per-dataset comparison of EvoFlows and baseline methods.* Comparison across multiple datasets (columns) and evaluation metrics (rows). The random transport model produces the most different sequences from the starting sequences as a function of the data. EvoFlows generates variants that remain close to the holdout distribution. Random baseline performs worst across all metrics. Evo-tuning without forced mutations achieves competitive scores but introduces almost no sequence mutations, as reflected in Levenshtein distances to x0, while forcing mutations significantly degrades performance across metrics.

Figure 7: *Per-position amino acid frequency heatmaps for all seed protein families.* Each row shows one dataset, with EvoFlows-generated sequences (left) and holdout sequences (right). Color intensity indicates per-position amino acid frequency after alignment. Conserved positions appear as points of high intensity, while variable regions show more diffuse patterns. The similar frequency profiles between generated and holdout sequences indicate that EvoFlows preserves the positional amino acid distribution characteristic of each homolog family.

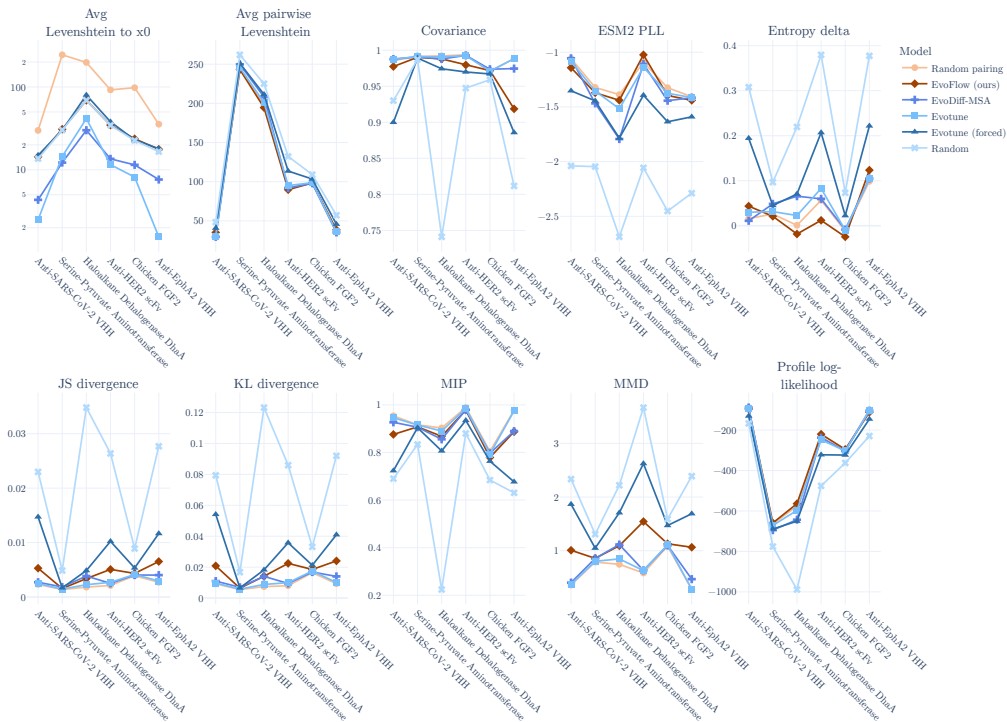

Figure 6: *Cross-dataset metric trends.* Each subplot shows one evaluation metric, with lines connecting per-dataset values for each method: Random pairing, EvoFlows (ours), Evo-tuning, Evo-tuning with forced mutations, and random baseline.

