# OpenReview forum: "EvoFlows: Evolutionary Edit-Based Flow-Matching for Protein Engineering"
_ICLR.cc/2026/Workshop/FM4Science — ICLR 2026 Workshop FM4Science Poster_

### Official Review · Reviewer_B4w4 · 2026-02-24
**Right Problem, Strong Fit, Awaiting the Indel Payoff**

**Rating:** 6
**Confidence:** 4

**Review:**

**Verdict.** This paper asks a well-posed question: can we model protein evolution as a learnable flow over substitutions, insertions, and deletions? The answer is promising. The method is sound, the writing is clear, and the appendix is thorough. The natural next milestone is validating indel generation on real proteins, which the authors acknowledge.

---

**Why this matters.** For readers less familiar with protein engineering: natural protein evolution operates through three elementary sequence operations, namely point substitutions (one amino acid replaced by another), insertions (new residues added), and deletions (existing residues removed). Substitutions are the most common and best studied. Insertions and deletions (collectively "indels") are rarer per event but important, as they reshape loop regions, alter binding interfaces, and drive the length variation that distinguishes antibody families. The CDR3 loop of antibodies, for instance, varies in length across clones, and this diversity is central to immune repertoire function.

Current protein language models handle only one of these operations natively. MLMs like ESM-2 predict substitutions at fixed positions but cannot change sequence length. AR models like ProGen2 generate full sequences left-to-right but lack the notion of locally editing a template. Diffusion models like EvoDiff produce variable-length sequences but do not model the edit path connecting ancestor to descendant. ESM-3 (Hayes et al., 2024) supports iterative masked generation and is now the most widely used generative PLM, but it also does not model explicit edit operations. In practice, engineers who need to introduce indels must resort to manual design or heuristics outside the model. EvoFlows is, to my knowledge, the first generative framework that supports all three operations in a single model, learning them jointly from evolutionary data. If this direction matures, it opens a design space that current tools cannot reach.

---

**What does this paper contribute?** The contribution is twofold. First, the paper identifies that the Edit Flows framework (Havasi et al., 2025), originally demonstrated on NLP tasks, has a natural fit in protein engineering. Protein evolution operates through elementary edits; homologous pairs from UniRef30 and OAS represent real evolutionary trajectories; Needleman-Wunsch alignment directly produces the extended-space representations the CTMC framework requires. The 21-token amino acid alphabet and typical sequence lengths of 100 to 500 residues also mean that several scalability concerns from the original Edit Flows work (vocabulary size, long-sequence generalization) are resolved by the domain itself.

Second, the paper provides initial empirical evidence. The deterministic benchmark (Table 1, Figure 4) shows the model learns position-dependent edit rules with F1 scores of 0.808 (insertion), 0.823 (substitution), and 0.879 (deletion), stable across lengths from 50 to 1000 residues. The confusion matrix shows remaining errors are primarily confusions with the no-op class rather than between edit types, which is a reasonable failure mode. On real proteins, the substitution-only evaluation across six seed families (Figure 3, Figure 5) shows EvoFlows matching Evotuned ESM-2 on distributional metrics (profile log-likelihood, covariance, MI with APC, BLOSUM-corrected KL, MMD) while producing more sequence diversity. The per-position amino acid frequency heatmaps in Figure 7 also show that generated sequences closely mirror the positional distributions of held-out natural homologs.

The architecture is clean: FiLM conditioning on a frozen ESM-2 trunk, three-headed MLP with separate rate and token predictions per edit type, and grid-free CTMC inference with clock normalization as a practical improvement over the original Edit Flows sampling.

---

**Indel evaluation on real proteins.** The paper states that "all benchmarks concern substitutions only" because no standard method exists for obtaining indels from MLMs, making direct comparison infeasible. This is fair, and the deterministic benchmark does confirm the model can learn and execute indels in a controlled setting, which serves as valid preliminary evidence.

There are evaluation strategies that do not require head-to-head baselines: reconstructing known evolutionary indels from MSA databases and measuring recovery accuracy, checking whether generated indels preserve predicted structural integrity via ESMFold, or benchmarking against protein evolution simulations with ground-truth indel trajectories. Controlled experiments like generating 100 to 101 residue insertions and comparing against natural length-variant homologs could also be informative. I expect this is the natural direction for the next iteration.

---

**On the results and baselines.** Looking at Figure 3 and Figure 5, the per-metric advantage of EvoFlows over Evotuned ESM-2 is real but modest. The main contribution is not outperforming Evotuning on individual metrics, but achieving comparable quality while generating more diverse edits. Evotuning without forced mutations barely modifies the input (as low as 1.5 edits per sequence on average); forcing mutations degrades quality. EvoFlows avoids this trade-off because edit operations are learned jointly rather than imposed post-hoc, and the paper could frame this point more explicitly.

The baseline comparison is limited to Evotuned ESM-2 variants and random mutations. The protein generative landscape is broader: ESM-3 can produce variants via iterative decoding, AR models like ProGen2 and ProtGPT2 exist, and EvoDiff handles variable-length discrete diffusion. ESM-3 in particular is not mentioned in the paper, even in Related Work. Implementing all of these may not be feasible at this stage, but discussing how EvoFlows relates to ESM-3's generation capabilities would strengthen the positioning. I am also curious how EvoFlows would compare against AR protein models on the same benchmarks.

---

**Minor notes.** Figure 1 gives a helpful overview of the edit process, but the diagram could benefit from a brief legend clarifying the color coding and arrow semantics, since it appears in the Introduction before the method is explained. Clock normalization controls the number of edits independently of sequence length; some guidance on setting it for new protein families would help practitioners. Publishing the implementation would be valuable, given that the original Edit Flows code from Meta remains unreleased. Reporting inference wall-clock times would also help contextualize the CTMC sampling cost.

---

**Recommendation: Borderline Accept.** The scientific question is well-posed, the domain fit is strong, and the substitution results hold up across diverse protein families. The writing is clear and the supplementary material is detailed, with thorough metric definitions in Appendix B and informative per-position heatmaps in Figure 7. The main gap is that indel generation, the paper's most distinctive capability, has not yet been validated on real proteins. For ongoing work this is an acceptable scope, but it is the clear next step. I hope to see this work continue to develop.

---

### Official Review · Reviewer_ZG5u · 2026-02-25
**This paper presents EvoFlows, an edit-based flow-matching model for protein design.**

**Rating:** 7
**Confidence:** 4

**Review:**

This paper presents EvoFlows, an edit-based flow-matching model for protein design. Unlike existing protein language models which estimate token likelihood or existing flow-matching models which learn to denoise corrupted inputs, EvoFlows uses edit flows to model the mutational trajectories between homologous protein sequences, including substitutions, insertions, and deletions.

Pros:

- The use of edit flows in the domain of protein is novel and suitable, as the edit operations naturally correspond to the basic operations in protein engineering, including insertions, deletions, and substitutions.

- EvoFlows novelly models the flow between homologous protein sequences, which in some sense simulates the protein engineering trajectories to improve protein functions.

- The benchmarking results show that EvoFlows can generate protein variants that are both natural-like while being diverse.

Cons:

- My main concern is that since the flow is modeling the trajectory between homologous protein sequences, there is not really a direction of function-improving or function-disrupting. Then, how would this EvoFlows framework be used in protein engineering, as claimed in the paper. The authors are suggested to discuss the application in protein engineering, where the goal is to improve protein function instead of generating natural-like sequences.

---

### Official Review · Reviewer_zntH · 2026-02-25
**Review of EvoFlows: Evolutionary Edit-Based Flow-Matching for Protein Engineering**

**Rating:** 5
**Confidence:** 3

**Review:**

The paper introduces EvoFlow, a variable-length sequence-to-sequence protein modeling approach focused on protein engineering using a discrete flow-matching approach to protein optimization to learn edit-based sequence-to-sequence transition rules.  Prior work involves training Protein Language Models (PLMs) using autoregressive models, which are inefficient, masked PLMs that require a fixed length, and discrete diffusion models that share similar challenges. EvoFlows leverages Discrete Flow Matching (DFM) and Edit Flows to enable variable-length sequence generation, does not rely on ad hoc mask insertions, and can modify sequences locally to enable protein optimization.

**Strengths:**
* The use of discrete flows to protein optimization is compelling, and the improvements over existing approaches to optimization with PLMs are well motivated
* The evaluation metrics are extensive, for example BLOSUM-corrected KL divergence is a thoughtful metric design choice, the diversity of protein families tested (enzymes, antibodies, growth factors) is a genuine strength, and the clock normalization hyperparameter giving controllable edit rates is a practical contribution
* Method derivations are explained in detail and are useful

**Weaknesses**
* Table 1 is only for EvoFlows, but doesn’t have any comparison with existing methods for the deterministic benchmark. It is hard to assess the efficacy of the deterministic benchmark without any comparison, even if the performance metrics are fairly high.
* Benchmarking is only done for substitutions for the 6 seed proteins. The authors acknowledge that no widely used method exists for obtaining indels, but the strength of the central claim on this enabling insertion, deletion, and substitutions is weakened or should be rephrased to “works on substitutions” and shows potential for other edits.
* The Evotuning baseline comparison, while informative, is a bit asymmetric as Evotuning wasn't designed for the edit-generation task EvoFlows targets. Comparing against EvoDiff (https://github.com/microsoft/evodiff) or similar diffusion-based protein generators would strengthen the experimental section
* The paper motivates EvoFlows partly by criticizing the inefficiency of autoregressive models for local optimization, yet provides no runtime or computational comparisons. Even a rough comparison of wall-clock time or number of forward passes needed per variant would help readers assess this practical advantage. (i.e. CTMC sampler vs MLM interactive infilling or diffusion denoising)


**Clarification:**
* The abstract highlights a 'grid-free inference procedure' as a contribution, but Section 3.3 doesn't use this terminology or clearly benchmark the practical advantages of this approach over the Euler integration it replaces. Could you better substantiate Section 3.3 to align with the abstract’s claim?
* Are ESM-2 encoder weights frozen or finetuned during EvoFlows training? If finetuned, by how much (learning rate, number of steps)?
* The abstract mentions an “improved ability” to generate non-trivial yet natural-like mutants, but can this be further substantiated (is this solely on the higher Levenshtein distance)?

**Overall Decision:**  5/10 Borderline Reject. The paper presents a well-motivated application of edit flows to protein engineering, and the core insight that evolutionary mutations map naturally to discrete edit operations is compelling. However, the evaluation does not sufficiently support the paper's central claims. The primary differentiator over masked language models: native support for insertions and deletions is only evaluated on a synthetic benchmark (Table 1) without baselines and is entirely absent from the natural protein experiments, which are substitution-only. The "improved ability" claim in the abstract reduces to higher Levenshtein distances at comparable distributional metrics, which is a reasonable but undersold argument that deserves explicit framing. Additionally, the efficiency motivation against autoregressive models is never quantified. The method itself is promising, and the mathematical framework is sound, but in its current form, the gap between what is claimed and what is demonstrated is large. Strengthening the indel evaluation, adding at least one computational cost comparison, and more precisely scoping the claims would substantially improve the paper and could revise to a weak accept.

---

### Official Review · Reviewer_Pab5 · 2026-02-25
**Review to Submission56**

**Rating:** 7
**Confidence:** 3

**Review:**

This paper presents EvoFlows, a protein language model based on edit flows (variable-length discrete flow matching). The model generates functionally conserved variants by performing substitution, insertion, and deletion operations on template sequences. The authors define pairing distributions over homologous proteins, employ the Needleman-Wunsch algorithm for pairing alignment, and train a CTMC rate model parameterised by an evolution-tuned ESM-2 encoder (combined with FiLM temporal conditioning). EvoFlows was evaluated across two scenarios: (i) synthetic deterministic editing benchmarks based on known true editing rules; (ii) six native protein families sourced from UniRef/OAS databases. Performance was compared against evolution-tuned masked language models and several baseline models using multiple sequence distribution metrics.

### Strengths
1. This paper demonstrates a genuine disparity between standard language models and the requirements of protein optimisation.
2. The synthetic experimental framework designed in Section 4.1 provides an exceptionally streamlined approach for evaluating edit-stream learning.
3. Evaluation criteria extend beyond ‘log-likelihood values for specific models’; multiple biologically meaningful statistical metrics are defined in the appendix.
4. Employing a frozen ESM-2 encoder coupled with a post-processing FiLM temporal attunement layer constitutes both a pragmatic and well-founded decision.

### Weaknesses
1. Evaluation is entirely in silico and focused on distributional similarity, not functional fitness.
2. The model training distribution and generalization story are somewhat narrow.

I am not familiar with Protein Engineering. For opinions within the relevant field, please refer more extensively to the views of other, more specialised reviewers.

---

### Official Review · Reviewer_7KmE · 2026-02-25
**EvoFlows: Evolutionary Edit-Based Flow-Matching for Protein Engineering**

**Rating:** 8
**Confidence:** 3

**Review:**

**Summary**

- The paper introduces EvoFlows, a novel protein language modeling framework tailored to protein engineering and local sequence optimization.
- Unlike the prior autoregressive and masked protein language models (PLMs), EvoFlows builds on a discrete edit-flow matching paradigm based on recent advances in flow matching with edit operations.
- Models protein evolution as elementary edit sequences of substitutions, insertions, and deletions between evolutionarily related (homologous) protein sequences
- Leveraging a discrete flow matching formulation to learn transition rates between aligned sequence pairs using a Bregman divergence loss.
- Defined sequence pairs from natural homologs with standard global alignment (Needleman–Wunsch) and train/test splits per protein family.
- On six diverse real protein families (enzymes, growth factors, antibody fragments), EvoFlows generates variants whose distribution better matches the held-out homologs compared to baselines. They use metrics such as Levenshtein distance, profile statistics, pseudo-log-likelihood, JS/KL divergence, and MMD.


**Strengths**

- Novel Generative Paradigm for Protein Sequence
- Joint Modeling of Edit Type and Position
- Deterministic Benchmark to Validate Edit Mechanisms:

**Weakness**

- Limited comparison against baselines
- Evaluation remains completely computational.

**Assessment**

Conceptually clear and well-written paper that advances generative modeling for protein engineering.

---

### Official Review · Reviewer_ygbx · 2026-02-25
**A sequence-to-sequence flow model that transforms one protein into another. Very interesting premise, but paper is hard to follow.**

**Rating:** 6
**Confidence:** 2

**Review:**

- The premise of this paper, a sequence-to-sequence flow model that transforms one protein into another through a series of substitutions, insertions and deletions, is extremely interesting with potential applications to drug discovery. The possibility of leveraging advances in language models for this purpose is exciting.
- Overall I feel that this paper is quite hard to follow unless you are an expert in both protein biology and flow models (of which I am neither). I would appreciate more pedagogical background section.
- The methods are thoroughly described, but writing a sentence for each step providing an intuitive description of what that step is doing would be very helpful.
- It is unclear to me exactly what the benefit is of editing a protein sequence into another one over just creating an entirely new protein sequence, other than computational time/efficiency. I understand that the methods outlined in this work ensure that the outcomes are "real" proteins, but presumably this is also the case with other methods? Is it that expressing it as a sequence of edits is more "natural" and relevant for actual protein editing?
- I am not sure what you mean by saying that autoregressive models "lack flexible controls in an optimization setting".
- While I appreciate writing out the equations, some mathematical terms are not defined, such as A* between Equations 5 and 6 and the delta function in Equation 7.
- The use of protein homolog databases to express the probability of transitioning between one protein sequence to another in terms of their evolutionary relationship is extremely interesting!
- There is very little interpretation of the results in Figure 3 outside of the caption, and I feel that this paper needs much more discussion of the metrics that are being used to evaluate the model and the model's performance compared to the baseline methods. From looking at Figure 3, it doesn't seem like EvoFlows performs better than some of the baseline models. If the point was that EvoFlows is more efficient, this should have been evaluated also.
- Overall I think this work is intriguing but the descriptions and explanations in the paper should be improved.

---

### Meta-Review · Area_Chair_oyqm · 2026-02-28

**Recommendation:** Accept (Poster)
**Confidence:** 3

**Metareview:**

EvoFlows is a novel sequence-to-sequence protein modeling approach that aims to tackle a highly challenging problem- handling variable-length sequences through the uses of discrete edits: insertions, deletion, and substitutions. The approach leverages edit flows between evolutionarily related protein sequences. Reviewers agreed that the method is timely and novel, and should be of interest to the workshop. It is recommended to improve the clarity of the paper, due to the extensive background required in both protein engineering and flow matching, to understand the method and appreciate its significance. Reviewers also recommend expanding the breadth and depth of the experimental validation, focusing on computational cost, indel edit effectiveness, and wet-lab validation.

---

### Decision · Program_Chairs · 2026-03-03

Accept (Poster)